

# Causal relationships between vegetation productivity, water availability, and atmospheric dryness at the catchment scale

Guta Wakbulcho Abeshu [1,2], Hong-Yi Li [2], Mingjie Shi [1], and L. Ruby Leung [1]

[1]Pacific Northwest National Laboratory, Richland, WA, USA
[2]Department of Civil and Environmental Engineering, University of Houston, Houston, TX, USA

**Correspondence:** Hong-Yi Li (hongyili.jadison@gmail.com )

**Abstract.** This study explores the causal relationships between catchment water availability, vapor pressure deficit, and gross primary productivity across 341 catchments in the contiguous US. Seasonal climatic, hydrological, and vegetation characteristics were represented using the Horton index, ecological aridity index, evaporative fraction index, and carbon uptake efficiency. Statistical methods, including circularity statistics, correlation analysis, and causality tests, were employed to determine the complex interactions between catchment wetness, atmospheric dryness, and vegetation carbon uptake. The results revealed a maximum lag of two months in the intra-annual variability of catchment water supply-productivity and atmospheric water demand-productivity relationships, with hysteresis patterns varying with the catchment's hydrological characteristics. In catchments not permanently under water-limited or energy-limited conditions, vegetation experiences hydrological stress during the peak growing period, coinciding with the highest gross primary productivity and carbon uptake efficiency being out of phase with Horton index and in phase with evaporative fraction index. Causality analysis highlights strong temporal continuity in GPP seasonal characteristics, with a cause-effect relationship between catchment water supply, atmospheric demand, and vegetation productivity spanning a maximum of two months. These findings underscore the need for a comprehensive functional framework that integrates catchment water supply, atmospheric demand, and vegetation productivity to enhance our understanding and predictive capabilities of ecosystem responses to climate change.

## 1 Introduction

Soil wetness and vapor pressure deficit (VPD) are two critical abiotic factors that limit ecosystem productivity and play vital roles in understanding vegetation carbon dynamics. Soil wetness determines the volume of water that plants can hydraulically lift (Gentine et al., 2019), while VPD controls the opening and closing of stomata (Grossiord et al., 2020). These factors are interconnected through the plant hydraulic transport system, which serves as a conduit between the processes at the leaf surfaces and the water supply at the roots. The structure and physiology of plants, including the stomata and hydraulic transport system, enable them to modulate their carbon assimilation rates in response to changes in soil wetness and VPD (Martínez-Vilalta et al., 2014). Understanding photosynthetic carbon assimilation in relation to soil wetness and VPD fluctuations is crucial for assessing the effects of climatic and hydrological processes on carbon dynamics in terrestrial ecosystem. However, understanding the distinct roles of soil moisture and VPD, along with their causal effects on vegetation carbon uptake, remains



a significant research challenge. This has led to an unharmonized representation of the importance of these two variables in data analysis and modeling experiments (Liu et al., 2020).

     The importance of soil wetness is particularly pronounced in water-limited subtropical ecosystems (Running et al., 2004; Seneviratne et al., 2010; Stocker et al., 2018), including drylands that make up about 41% of the global land surface (Cherlet et al., 2018). In these ecosystems, plant survival is primarily dictated by soil water availability (Anderegg et al., 2015), and the

nonlinear response of carbon fluxes to soil water variability is a critical process that affects an ecosystem's long-term capacity as a carbon sink (Green et al., 2019). Reduced soil wetness can increase near-surface temperature and decrease latent heat flux, which, under drought conditions, can result in extreme atmospheric aridity owing to soil wetness feedback (Zhou et al., 2019). A recent global analysis showed that soil wetness drives dryness stress on ecosystem productivity in over 70% of vegetated land areas, highlighting its significance in carbon dynamics and land-atmosphere interactions (Liu et al., 2020).

On the other hand, VPD is a crucial driver of plant function and a determinant of plant-water relations, in some instances influencing vegetation carbon-water exchange more than soil water availability (Giardina et al., 2018; Novick et al., 2016). An increase in VPD augments the atmospheric demand for water, thereby influencing leaf stomatal conductance and latent heat flux. However, owing to plant regulatory mechanisms, this does not necessarily lead to decreased vegetation growth (Massmann et al., 2019; Yuan et al., 2019). High or rapidly increasing VPD causes plants to close their stomata, minimizing

water loss and preventing hydraulic transport system failure, even though it suppresses the photosynthetic rate (Grossiord et al., 2020; McAdam and Brodribb, 2015). With the rise in global temperatures and an expected increase in future VPD (Byrne and O'Gorman, 2013; Hatfield and Prueger, 2015), it is vital to quantify the impact of VPD on ecosystem productivity under both water-stressed and saturated conditions.

     Low soil wetness, high VPD, or a combination of both often triggers hydrological stress in vegetation (Fang et al., 2021;

Grossiord et al., 2020; Liu et al., 2020). Enhanced VPD, in combination with low soil wetness, can induce severe drought events (Zhou et al., 2019). Prolonged periods of such conditions can damage the plant's hydraulic transport system of plants, potentially increasing mortality rates. However, determining the specific soil wetness and VPD thresholds and their combined effect that precipitates hydrological stress presents a significant challenge (Fu et al., 2022). Conversely, a season with favorable conditions can stimulate vegetation growth and increase water usage, thereby accelerating the rate of soil wetness depletion. If

soil wetness conditions are unfavorable in the subsequent season, this can intensify the hydrological stress (Bastos et al., 2020). The phenomenon known as ecosystem structural overshoot often occurs when a preceding period of unusually large biomass leads to a supply-demand imbalance for the current period (Jump et al., 2017; Zhang et al., 2021). Many studies have indicated that structural overshoots significantly exacerbate drought events (Bastos et al., 2020; Buermann et al., 2018; Goulden and Bales, 2019; Wolf et al., 2012). Globally, reports indicate that structural overshoots were responsible for approximately 11%

of the drought events from 1981 to 2015 (Zhang et al., 2021). Understanding the lag in vegetation response to alterations in soil wetness and VPD is integral to a better understanding of these issues. This enabled us to anticipate and mitigate shifts in vegetation health and vitality due to changing climatic conditions, given the delayed reaction of vegetation to such changes rather than an instantaneous reaction.



In this study, we aimed to advance our understanding of the complex interactions among water availability, atmospheric
dryness, and vegetation productivity by investigating the response of ecosystem carbon uptake to intra-annual variability in
soil wetness and VPD at the catchment scale. To address the limitations of previous studies, we considered total soil wetness
instead of soil moisture across different soil layers to provide a more comprehensive representation of the water available
for plant use (Abeshu and Li, 2021). Furthermore, we use a 30-meter resolution gross primary productivity (GPP) product
to effectively capture the spatial heterogeneity of catchment GPP. Our study seeks to answer three key questions: 1) How
does catchment GPP respond to soil wetness and VPD across different vegetation types? 2) What are the critical features
are responsible for between-catchment differences in the vegetation responses? 3) How robust are the causal links between
these variables? By addressing these questions, we aim to provide valuable insights into the complex dynamics of ecosystem
productivity under varying hydrological and atmospheric conditions, with potential implications for ecosystem management
and climate change adaptation. The remainder of this paper is structured as follows: Sections 2 and 3 introduce the data and
methods, section 4 presents the results, section 5 discusses the findings, and section 6 concludes the paper.

## 2   Data

This study utilizes the CAMELS (Catchment Attributes and MEteorology for Large-sample Studies) dataset, featuring data
from 671 unimpaired catchments across the contiguous United States (Newman et al., 2015). The use of unimpaired catch-
ments allows for the analysis of vegetation response to soil wetness and VPD under natural conditions, minimizing the influence
of human interventions such as land use change or water management practices. These catchments span across various climatic,
topographic, and vegetation gradients, providing diverse samples for understanding the relationship between water availability,
atmospheric dryness, and vegetation productivity. The CAMELS dataset comprises daily observed and observation-based hy-
drometeorological datasets, including model outputs such as actual evapotranspiration (ET) from the integrated Snow-17/SAC-
SMA model (Addor et al., 2017; Newman et al., 2015). The CAMELS dataset also provides information on catchment at-
tributes, such as dominant vegetation cover characteristics. Our analysis depends on daily data, including precipitation (rain
+ melt), maximum and minimum temperature, actual vapor pressure, as well as actual and potential evapotranspiration (PET)
and stream discharge. The daily potential evapotranspiration is estimated using the Priestly-Taylor method. Baseflow is derived
from observed discharge using a one-parameter recursive digital filter with three passes (Nathan and McMahon, 1990).

Beyond the hydrometeorological data, this study also incorporates gross primary productivity data sourced from the Landsat
GPP dataset for the contiguous United States (Robinson et al., 2018). This dataset features a spatial resolution of 30 meters
and a temporal resolution of 16 days. The high spatial resolution of the Landsat GPP dataset is crucial for capturing the spatial
heterogeneity of catchment GPP. Accessible via Google Earth Engine, this data was masked using catchment polygons over
the period from 1986 to 2021. Subsequently, a time series of the average catchment GPP was constructed at 16-day intervals
and later transformed into a monthly series. The leaf area index (LAI) data was generated for each catchment from the AVHRR
dataset (Claverie and Vermote, 2014) and is used to characterize the vegetation density and growth stage in the catchments.
Quality control of the data is conducted based on two criteria. The first criterion is the complete absence of missing data



in both the model output and observation data. The second criterion is that the relative percent error between the simulated annual mean of the model output ET and the observed ET (calculated as the annual mean precipitation minus the annual mean discharge) must be less than ten percent. Adhering to these criteria results in a study period ranging from 1986 to 2014 and includes 341 catchments distributed across the contiguous United States (Fig. 1). These 341 catchments are divided into six vegetation groups: Deciduous Broadleaf (DBF) with 85 catchments, Evergreen Forest (Needle leaf + Broadleaf) or EF with 21 catchments, Mixed Forests (MF) with 40 catchments, Croplands or Croplands/Natural Vegetation Mosaic (CL/NVM) with 100 catchments, Grasslands (GL) with 43 catchments, and Savannas, Woody Savannas, or Open/Closed Shrublands dominated catchments (WSSL) with 51 catchments.

## 3   Methods

### 3.1   Water available for vegetation use

Storage carryover significantly modifies precipitation partitioning at both annual and sub-annual scales. By considering the inputs and outputs that impact the dynamics of surface and subsurface storage within catchments, the water balance at a monthly scale is expressed as:

$$P - ET - Q_b - Q_s = \Delta S \tag{1}$$

$$W - \Delta S = ET + Q_b \tag{2}$$

P is precipitation, ET is actual evapotranspiration, $Q_b$ is baseflow, $Q_s$ is surface runoff, and $\Delta S$ is the net change in water storage. The term $\Delta S$ encompasses changes in surface water storage (including streams, lakes, swamps, and surface depressions) and subsurface storage. Total wetting (W) refers to precipitation that wets the catchment, excluding precipitation that becomes surface runoff. It includes the precipitation that infiltrates and the portion stored on the land surface (i.e., rivers, lakes, swamps, and surface depressions); thus, $P = W + Q_s$. This represents the first stage of hydrologic partitioning. By substituting $P - Q_s$ as W in Equation (1), Equation (2) is derived, illustrating the second stage of hydrologic partitioning. Catchment wetness ($W - \Delta S$), representing the total water available for vegetation use, will henceforth be referred to as 'Wetness' throughout the manuscript. Understanding the dynamics of catchment wetness is crucial for assessing the impact of water availability on vegetation productivity and carbon uptake, which is a key focus of this study.

### 3.1.1   Catchment atmospheric dryness

VPD, which measures the extent of atmospheric dryness, is calculated as the difference between the actual vapor pressure (AVP) and saturation vapor pressure (SVP). The mean daily AVP was sourced from the CAMELS dataset and calculated the mean daily SVP using the Tetens formula, typically used in potential evapotranspiration (Allen et al., 1998). The mean daily SVP is the mean of SVP at maximum and minimum daily air temperatures, which is later converted to monthly. VPD is an important measure of atmospheric dryness, as it directly influences the water demand on vegetation and the rate of





evapotranspiration. Higher VPD values indicate a greater atmospheric moisture deficit, which can lead to increased water stress on plants. Investigating the relationship between VPD and vegetation productivity is crucial for understanding the impact of
125 atmospheric dryness on ecosystem carbon uptake, which is a focus of this study.

### 3.1.2 Catchment hydroclimatic and vegetation dynamics

For a more comprehensive understanding of catchment ecohydrological systems functionality, which is crucial for assessing the impact of water availability and atmospheric dryness on vegetation productivity, we incorporate indices based on ecological, hydrological, and energy perspectives. To evaluate the seasonal variation of climatic demand-supply interactions, we utilize
the Ecological Aridity Index (EAI). The Horton Index (HI) assesses the hydrologic demand-supply interaction, while the energy demand-supply state is characterized using the evaporation fraction (EFI). The EAI, calculated as the ratio of potential evapotranspiration to catchment wetness (Abeshu and Li, 2021), illustrates the interplay between catchment energy and water supply for plant water use. Its magnitude can vary from 0 to infinity, with a wetter climate corresponding to lower values. The HI is defined as evapotranspiration in proportion to the water available for vegetation use within the catchment (Abeshu and Li,
2021). HI can range from 0 to 1, indicating absolute hydrologic wetness and dryness conditions, respectively. EFI represents the actual to potential evapotranspiration ratio, indicating the catchment's energy use efficiency. Its magnitude ranges between 1 and 0, with 1 denoting the most efficient catchments and 0 indicating the least efficient ones. Furthermore, we use the Carbon Uptake Efficiency (CUE) to characterize catchment vegetation dynamics. According to the light use efficiency model GPP (Robinson et al., 2018; Jiang et al., 2021) is parameterized as,

$$140 \quad \text{GPP} = \varepsilon_{\text{max}} \cdot (T_{\text{scalar}} \cdot W_{\text{scalar}}) \cdot \text{APAR} \tag{3}$$

Where $\varepsilon_{\text{max}}$ is the maximum radiation conversion efficiency (kg°C MJ$^{-1}$) specific to a vegetation type, which is down-regulated by temperature limitation ($T_{\text{scalar}}$) and water stress ($W_{\text{scalar}}$) to yield actual radiation conversion efficiency, $\varepsilon = \varepsilon_{\text{max}} \cdot T_{\text{scalar}} \cdot W_{\text{scalar}}$, and APAR is the absorbed photosynthetically active radiation. Both $T_{\text{scalar}}$ and $W_{\text{scalar}}$ reflect the climatic limits of plant carbon uptake. Hence, under no limiting conditions (i.e., $T_{\text{scalar}} = W_{\text{scalar}} = 1$), Eqn. (3) leads to estimates
of potential GPP as follows:

$$\text{GPP}_{\text{potential}} = \varepsilon_{\text{max}} \cdot \text{APAR} \tag{4}$$

The ratio of actual to potential GPP, the CUE, can be expressed as:

$$\text{CUE} = \frac{\text{GPP}}{\text{GPP}_{\text{potential}}} = T_{\text{scalar}} \cdot W_{\text{scalar}} \tag{5}$$

CUE ranges between 0 and 1. The mean monthly $T_{\text{scalar}}$ and $W_{\text{scalar}}$ were estimated from the mean daily temperature and
150 VPD data along with Biome-Property-Look-Up-Table (Robinson et al., 2018). CUE = 1 represents an efficient catchment.

### 3.2 Statistical analysis

To comprehensively analyze the relationships between catchment hydroclimatic variables and vegetation dynamics, we employ a range of statistical methods tailored to our study's objectives. Circularity statistics are used to summarize the intra-annual



variability of fluxes, providing insights into the seasonality and timing of GPP, Wetness, and VPD. The Granger causality
test and PCMCI+ are employed to investigate potential causal relationships between these variables, helping to identify the
directionality and strength of their interactions. Principal Component Analysis (PCA) is utilized to explore the degree to which
long-term catchment characteristics explain the variability of mean monthly GPP−Wetness and GPP−VPD relationships,
aiding in the identification of key factors influencing these interactions. Finally, Pearson's correlation is used to quantify the
strength and direction of monotonic relationships between paired data. By applying these diverse statistical techniques, we aim
to gain a comprehensive understanding of the complex interactions between catchment hydroclimatic variables and vegetation
dynamics, and their implications for ecosystem functioning and carbon uptake.

**Circularity statistics:** Circular (directional) statistics is used to summarize the intra-annual variability of fluxes (Dingman,
2015; Fisher, 1993; Markham, 1970). We first convert the average monthly data into vector quantities to implement these
statistics. The vector's magnitude corresponds to the month's flux amount, and the vector direction ($\phi$) is the month expressed
in a unit of arc. The direction of a given month is the median date of the month measured from January 1st in a clockwise
direction. In a standard year with 365 days, one day equals $\frac{360}{365} = 0.986°$ on a circle. This factor adjusts the day of the year
to give the corresponding angular direction on a circle. The mean monthly vector components ($C$ and $S$) of any catchment
flux $F_m$ were determined as $C = F_m \cos\phi_m$ and $S = F_m \sin\phi_m$. The resultant, $R$, is the square root of the sum of the squares
of $C$ and $S$ (i.e., $\sqrt{\sum C^2 + \sum S^2}$). The Seasonality Index (SI), a measure of the degree of variation of a given catchment
flux throughout the year (Fisher, 1993), was obtained by dividing the resultant vector $R$ by the annual mean flux. SI values
range from 0 to 1. A value of 0 suggests a flux uniformly distributed intra-annually, while 1 indicates a flux concentrated
within a single month. The average time of occurrence ($\overline{\phi}$) corresponds to the angular direction of the resultant vector. In this
framework, a $\overline{\phi}$ for January 1st represents the north (0°), April 1st represents the east (90°), July 1st represents the south (180°),
and October 1st represents the west (270°). Utilizing this framework, we computed the seasonality index and average time of
175 occurrence for GPP (SI$_{gpp}$ and $\phi_{gpp}$), Wetness (SI$_{wetness}$ and $\phi_{wetness}$), and VPD (SI$_{vpd}$ and $\phi_{vpd}$) for all catchments. Note that
the time of occurrence estimation from circularity statistics is less meaningful when the seasonality is very weak.

**Granger causality test:** Granger causality is a statistical concept used to determine if one time series can help predict another
(Stokes and Purdon, 2017). The test is based on the principles of temporal precedence and predictability (Granger, 1969). That
is, if one time series causes another, then past values of the causing series should contain information that can be used to
180 improve the prediction of the second series (Stokes and Purdon, 2017). The Granger causality test involves regressing each
time series on its own past values and the past values of the other series. If the coefficients are significant, the test concludes
that the first-series Granger causes the second series. Note that the Granger causality test does not prove true causality in
the philosophical sense. It only shows that one series can be used to forecast another, not whether changes in the first series
necessarily cause changes in the second.

**PCMCI+ for causal analysis:** PCMCI+ is a statistical method to discover potential causal relationships between time
series. It blends two key components: the Peter and Clark (PC) algorithm and the momentary conditional independence (MCI).
The PC algorithm, a constraint-based method for causal discovery, is used to select conditions, while MCI, a measure of
the degree to which two random variables are independent given the values of other variables, is used to test for momentary





conditional independence (Runge, 2018; Runge et al., 2019a). The underlying assumption of PCMCI+ is that if two variables demonstrate statistical dependence, they may hold a causal relationship, and conversely, if they are statistically independent, likely, they do not have a causal relationship (Runge et al., 2019a, b). It is crucial to remember that while PCMCI+ can suggest causal relationships, it does not confirm them. PCMCI+ has been tested and applied to flux tower data (Krich et al., 2022, 2020), and we apply it to the catchment scale to discover the GPP−Wetness and GPP−VPD causal links. A partial correlation is employed for conditional independence test statistics to assess the causal strength.

**Principal Component Analysis (PCA):** PCA is a statistical technique commonly used in data analysis and machine learning. It is a dimension reduction method that transforms a large set of variables that may be correlated into a smaller set of uncorrelated variables called principal components. The first principal component accounts for as much of the variability in the data as possible, and each succeeding component accounts for as much of the remaining variability as possible under the constraint that it is orthogonal (uncorrelated) to the preceding components. PCA identifies the axes in the data space along which the data varies the most and reorients the data along these axes. This process of transformation and reduction can help simplify the data description and highlight important relationships between variables. We employed PCA to explore the degree to which long-term catchment characteristics explain the variability of mean monthly GPP−Wetness and GPP−VPD relationships.

**Pearson's r:** Pearson's correlation is a metric for quantifying the degree of a monotonic relationship between paired data. It ranges from $-1$ to $+1$. Generally, $0 < |r| \leq 0.20$ is considered negligible, $0.21 < |r| \leq 0.40$ is weak, $0.41 < |r| \leq 0.60$ is moderate, $0.61 < |r| \leq 0.80$ is strong, and $0.81 < |r| \leq 1.00$ is very strong.

## 4 Results

We evaluated the strength of monotonic relationship strength between the three components (i.e., GPP, VPD, and Wetness) at annual and monthly scales using Pearson's r. In 72% of the study catchments, we observed a strong negative correlation between Wetness and VPD on an annual scale. Another 20% of the catchments exhibited a moderate negative correlation. For the monthly scale, after grouping the data by month, we computed the correlation coefficient between Wetness and VPD for each month. During months of high water demand (June-August), we found a moderate to strong negative correlation in 80% of the catchments (Fig. 2a). This pattern persisted for 73% of the catchments in September. We carried out a similar monthly scale analysis for GPP−VPD and GPP−Wetness (Figs. 2b and c). Over 60% of the catchments demonstrated a moderate to strong positive correlation between Wetness and GPP during the peak growing months (June-August). A moderate to strong negative correlation between GPP and VPD emerged in more than 66% of the catchments from June to September. During the most productive months (June-August), a weak correlation of GPP−Wetness and GPP−VPD persists for 30-40% of the catchments (Fig. 2b and c). This could be because there is a lag between the vegetation's response to water supply and demand. To investigate this, we performed cross-correlation analyses for GPP−Wetness and GPP−VPD using monthly data. We found that the best association between GPP and Wetness is at zero lag (i.e., vegetation responds to a change in water supply in the same month) for 57% of the catchments and at one-month lag (i.e., vegetation responds to a change in the water supply after one month) for another 37%. The correlation coefficient at the corresponding lags is $\geq 0.8$ for all catchments. Similarly, the





best association between GPP and VPD is at zero lag for 14.5% of the catchments and a one-month lag for 73%. The results suggest that water supply-productivity and water demand-productivity cause-effect interactions occur within a maximum span of two months (i.e., +1 month from GPP). Granger causality tests indicated that Wetness and VPD significantly affected GPP in all catchments, demonstrating their compound effect on seasonal GPP patterns.

We further explored the spatial relationships among GPP, Wetness, and VPD across different vegetation types using mean monthly values. We conducted these analyses independently for each of the six vegetation classes described in Section 2. Our results revealed a strong positive association ($r \geq 0.61$) between GPP and Wetness in WSSL and GL catchments (Fig. 3a), which is consistent with the expectation that these vegetation types, inhabiting water-limited environments, would exhibit rapid responses to changes in water availability. For CL/NVM and DBF, the relationship ranged from moderate to strong and positive, except during the peak carbon uptake period in the summer months (June-August). In contrast, EF (4 months) and MF (5 months) showed a moderate relationship ($r > 0.41$) only during the dormant months (October-March) (Fig. 3a). The GPP−VPD relationship exhibited a distinct seasonal pattern across vegetation types (Fig. 3b). During the dormant months (typically October-March), we observed a moderate to strong positive relationship ($r \geq 0.41$) for all vegetation types, except for WSSL. Conversely, during the peak growing season (June-August), the relationship was moderate to strong and negative ($r \leq -0.41$). This negative association can be attributed to the relatively high atmospheric water demand during these months, which tends to induce stomatal closure in plants, reducing carbon uptake relative to the potential. The positive association between GPP and Wetness for most vegetation types during the non-growing periods suggests a relatively rapid vegetation response to changes in catchment water supply. However, the lack of a significant GPP−Wetness association during the most productive months, except for WSSL and GL, coupled with a strong negative GPP−VPD association, implies a delayed response to catchment water supply in most catchments during this period. These findings highlight the complex interplay between water availability, atmospheric dryness, and vegetation productivity across different ecosystems. The varying strengths and directions of the relationships between GPP, Wetness, and VPD demonstrate the importance of considering both the spatial and temporal dimensions when investigating the drivers of ecosystem productivity. Understanding these relationships is crucial for predicting the responses of different vegetation types to changes in water availability and atmospheric dryness, with implications for ecosystem functioning and carbon uptake in the face of climate change.

To better understand how the GPP−Wetness and GPP−VPD relationships change throughout the year, we used circular statistics to summarize their intra-annual variability. This analysis yielded two statistical measures for each variable: Seasonality Index (SI) and average time of occurrence ($\phi$). The SI values varied with geographic latitude, with a general trend of increasing SI from south to north for all three variables within a given longitudinal swath (Fig. 4a). Comparing the SI values among the variables revealed that $\mathrm{SI_{gpp}} > \mathrm{SI_{wetness}}$ for 86% of the catchments and $\mathrm{SI_{gpp}} > \mathrm{SI_{vpd}}$ for 92% of the catchments, indicating that catchment vegetation productivity exhibits greater intra-annual variability than both catchment wetness and atmospheric demand. Furthermore, $\mathrm{SI_{wetness}} > \mathrm{SI_{vpd}}$ for 66% of the catchments, suggesting that atmospheric water demand is the least varied component among the three in most cases. When converting the angular estimations of $\phi$ to months, we found that the average time of occurrence for Wetness and GPP matched for 73% of the catchments, while $\phi_{\mathrm{gpp}}$ was delayed by at least one month for another 23% (Fig. 4b). The $\phi_{\mathrm{vpd}}$ differed by at least +1 month from $\phi_{\mathrm{gpp}}$ and $\phi_{\mathrm{wetness}}$ for 91% and 95%





of the catchments, respectively. However, it is important to note that the time of occurrence is less meaningful when the GPP seasonality is weak; therefore, we relied primarily on the seasonality strength for further analysis. These findings highlight the spatial variability in the seasonality of GPP, Wetness, and VPD across the study catchments (Fig. 4c). The higher SI values
for GPP compared to Wetness and VPD suggest that vegetation productivity is more sensitive to seasonal changes than water availability and atmospheric dryness. The differences in the timing of peak values for GPP, Wetness, and VPD, as indicated by the $\phi$ values, further underscore the complex interplay between these variables and the potential for lagged responses of vegetation to changes in water supply and demand. The strength of seasonality, combined with lag in vegetation response, creates hysteresis between GPP and the abiotic driving variables, namely Wetness and VPD. Hysteresis is a phenomenon that occurs
when changes in an effect lag behind changes in the causal variable. We first examined the hysteresis patterns of GPP$-$Wetness and GPP$-$VPD in the six dominant vegetation groups. To standardize the comparison across catchments within each group, we normalized all three variables by their mean values exceeding the 90th percentile. Figures 5 and 6 illustrate the GPP$-$Wetness and GPP$-$VPD hysteresis in the six vegetation groups. As displayed, hysteresis can manifest in various defining patterns such as size and direction. Hysteresis can be narrow (e.g., Fig. 5e) or wide (Fig. 5a) based on size, and it can proceed in a clockwise
(e.g., Fig. 6) or counterclockwise direction (i.e., Fig. 5). The lag between the variables primarily dictates the direction of the hysteresis, whereas factors influencing the size of the hysteresis can differ for GPP$-$Wetness and GPP$-$VPD hysteresis.

To establish a standard measure of the relative size of the hysteresis loop for comparisons across catchments, we calculated the area within the loop. We probed the drivers of these characteristics by assessing the relationships between the hysteresis loop area and the long-term catchment characteristics using PCA. First, we evaluated several variables in relation to the areas of the
hysteresis loops, filtering out those that showed a significant correlation with both the GPP$-$Wetness and GPP$-$VPD hysteresis loop area. The identified variables include the long-term climatic aridity (with $\rho_{\mathrm{GPP-Wetness}} = -0.381$, $\rho_{\mathrm{GPP-VPD}} = -0.677$), PET-P phase-index ($\rho(PET, P)$, phase agreement between P and PET seasonal pattern) ($\rho_{\mathrm{GPP-Wetness}} = -0.242$, $\rho_{\mathrm{GPP-VPD}} = -0.379$), peak LAI ($\rho_{\mathrm{GPP-Wetness}} = 0.531$, $\rho_{\mathrm{GPP-VPD}} = 0.707$), the fraction of forest ($\rho_{\mathrm{GPP-Wetness}} = 0.482$, $\rho_{\mathrm{GPP-VPD}} = 0.674$) and vegetation root depth ($\rho_{\mathrm{GPP-Wetness}} = 0.511$, $\rho_{\mathrm{GPP-VPD}} = 0.62$). We then conducted a PCA on these variables in relation to
the area of the hysteresis loop, the results of which are depicted in Fig. 7. The first two components from the PCA collectively accounted for more than 80% of the variability in the loop sizes for both the GPP$-$Wetness and GPP$-$VPD hysteresis. For both GPP$-$Wetness and GPP$-$VPD, all variables, except for the PET-P phase-index, made significant contributions to the variability along the first principal component (PC-1), as shown in Fig. 7b. However, the PET-P phase-index was the dominant contributor to the variability along the second principal component (PC-2), but only for GPP$-$Wetness, as illustrated in Fig.
7c.

The intra-annual variability within individual catchments revealed two primary patterns in the relationships between GPP and the abiotic drivers, Wetness and VPD. These patterns manifest in the size and direction of the hysteresis loops, as depicted in Figure 8. Firstly, regarding the direction of hysteresis, VPD typically peaked approximately one month after the GPP peak, creating a clockwise hysteresis loop when GPP is plotted as a function of VPD (Figs. 8c and 8f). In contrast, for the majority
of the 341 catchments analyzed, the intra-annual Wetness peak precedes or coincides with the GPP peak, resulting in a counterclockwise hysteresis loop when GPP is plotted against Wetness (Fig. 8b). However, a relatively clear clockwise pattern is





observed in 40 of these catchments (Fig. 8b), characterized by low seasonality in both the HI and the EFI, with high values throughout the year. Secondly, the size of the hysteresis loop varies across catchments. The catchments exhibiting a clockwise GPP−Wetness hysteresis pattern also display low seasonality in the carbon uptake efficiency and low monthly CUE values, resulting in a narrow hysteresis loop. These findings suggest that a narrow hysteresis predominantly occurs when Wetness approaches the PET across all months, indicating a minimal lag between GPP and Wetness.

Figure 9 presents the causal strengths between monthly GPP, its past values, and its relationship with Wetness and VPD, considering lags of up to four months. Our analysis uncovers a strong positive causal link in GPP autocorrelation at a one-month lag across diverse catchments (Fig. 9d). This finding suggests that a given month's GPP value is significantly influenced by the preceding month's value, echoing the temporal continuity frequently observed in biological and environmental time series. However, this positive correlation inverts to negative at a two-month lag (Fig. 9g). This unexpected pattern, persisting in catchments where the causal link remains statistically significant, is more likely to indicate a spurious connection rather than natural ecological processes. We hypothesize that the seasonality typical of environmental data could be the source of such anomalies, potentially introducing misleading correlations. As a result, we consider only the one-month lag as a valid connection in our analysis. The strong positive autocorrelation in GPP at a one-month lag (Fig. 9d) suggests that vegetation productivity in a given month is significantly influenced by the conditions and dynamics of the preceding month. This temporal carry-over effect could arise from various factors, such as the persistence of environmental conditions (e.g., soil moisture, temperature) or the lagged response of vegetation to changes in these conditions due to physiological processes like carbon allocation and biomass accumulation. Capturing this autocorrelation is crucial for accurately representing the inertia and memory effects in ecosystem processes and improving the predictive capabilities of vegetation productivity models.

Causal links between Wetness and GPP generally exhibit a positive and statistically significant relationship in 99% of the catchments at zero lag (Fig. 9b). The proportion of catchments with a significant positive connection reduces to 81% at a one-month lag (Fig. 9e), 34% at a two-month lag (Fig. 9h), and 5% at a three-month lag (Fig. 9k). As observed in the GPP autocorrelation, negative MCI values are regarded as spurious, primarily due to our expectation of a positive influence of catchment water availability on vegetation productivity. The strength of the VPD-GPP causal links is only significant for 194 catchments at lag zero (Fig. 9c), of which 153 are positive and 41 are negative. These results display a spatial pattern: catchments with a negative MCI are predominantly in arid regions, while those with a positive MCI are generally found in relatively humid regions. The contrasting spatial patterns observed for the VPD-GPP causal links at lag zero (Fig. 9c) highlight the varying responses of vegetation productivity to vapor pressure deficit (VPD) across different hydroclimatic regimes. In arid regions, characterized by low water availability, high VPD levels can induce stomatal closure in plants as a water conservation mechanism, leading to a negative causal link between VPD and GPP. Conversely, in humid regions with abundant water supply, moderate VPD levels can stimulate transpiration and carbon uptake, resulting in a positive causal link. These divergent responses reflect the intricate balance between water demand and supply, as well as the adaptations of vegetation to their respective environmental conditions. Incorporating this spatial variability in the VPD-GPP relationship is crucial for accurately representing the coupled water and carbon cycles in terrestrial ecosystem models, particularly under changing climatic conditions. Consequently, a positive causal link is prevalent in humid climates, while a negative causal link is observed in arid





climates. At a one-month lag, approximately 80% of the 341 catchments demonstrate a positive causal link, albeit with varying

degrees of strength (Fig. 9f). Both the number of catchments exhibiting significant causal links and the strength of these links

decrease as the lag increases (Figs. 9i and l).

## 5   Discussions

Our analysis of Wetness-VPD relationships at the annual scale revealed that wet and dry years correspond to low and high

atmospheric water demands, respectively. This finding aligns with previous research that investigated the relationship between

annual soil wetness and VPD (Liu et al., 2020; Seneviratne et al., 2010; Zhou et al., 2019). Delving deeper, our monthly scale

analysis showed a robust negative correlation between Wetness and VPD during the most productive periods for vegetation

(i.e., June, July, and August) while during other months, this negative correlation was observed in fewer than 25% of the

341 catchments we analyzed. This suggests that the critical productive months may disproportionately influence the patterns

observed at the annual scale. Our lag correlation analysis between GPP, Wetness, and VPD hints at a complex supply-demand-

productivity cause-and-effect process at the catchment scale, typically unfolding over a span of two months. The GPP responds

to Wetness with a maximum lag of one month, whereas VPD generally lags behind GPP by one month. Previous research has

shown similar characteristics in the GPP−VPD relationship, even for diurnal scale analysis (Zhou et al., 2014).

The vegetation's response time is the primary determinant of the hysteresis direction between GPP−Wetness, as well as

GPP−VPD. As water availability is a crucial driver for plant growth, a delay in response typically results in a counterclockwise

GPP−Wetness hysteresis curve. Conversely, a clockwise hysteresis curve is somewhat unexpected. Catchments that show this

feature usually have high energy and water-use efficiency yet low carbon uptake efficiency throughout the year. Interestingly,

most months for these catchments show that ET approximates Wetness, meaning ET predominantly defines the second stage

of hydrologic partitioning, and the hysteresis between GPP and ET follows a clockwise direction, as affirmed by prior research

(Zhou et al., 2014). Catchments with a counterclockwise GPP−Wetness hysteresis exhibit three unique traits: i) HI and EFI

are not synchronized, ii) CUE aligns with HI during the greening and browning phases, and iii) CUE is in-phase with EFI

during peak growing periods. It is worth noting that the mismatch between CUE and HI during peak carbon uptake periods

in most catchments indicates that their vegetation experiences hydrologic stress. Catchment increases its water use efficiency

as it progressively dries out. In other words, the amount of carbon taken up by vegetation at the expense of one unit of water

increases with catchment hydrological stress (i.e., ↑ HI). This is validated seasonally by a strong correlation between HI

and GPP. However, there is a decrease in carbon uptake efficiency (i.e., the ratio of carbon absorbed to its potential value)

with increasing HI, especially during periods of hydrologic stress (i.e., when HI → 1). This pattern holds for catchments that

remain water-limited throughout the year. For catchments where hydrologic conditions alternate between water-limited and

energy-limited states within a year, this phenomenon occurs specifically when the catchment is in a water-limited state. Both

intra-annual and long-term hydrologic variations strongly correlate with the size of the hysteresis loop. Wetter catchments

typically display wider hysteresis, while narrow hysteresis is typical in dry catchments. A narrow hysteresis curve signals a

catchment that is efficient in energy and water use but falls short in carbon uptake relative to its potential. Overall, the hysteresis





loop size variation between catchments can be sufficiently explained by climatic (e.g., aridity, sync in the phase of PET and P) and landscape features (e.g., fraction forest, LAI, and root depth).

Our causality analysis reveals significant influences of GPP (autocorrelation), Wetness, and VPD on the current month's GPP at zero and one-month lag. The strong one-month autocorrelation of GPP across different catchments reinforces the prevalent notion of temporal continuity in environmental processes. The negative correlation observed at a two-month lag, which we
initially perceived as counterintuitive, invites further scrutiny. The inherent seasonality in environmental datasets might explain these anomalies, suggesting careful interpretation. This further underlines the need to differentiate natural ecological processes from statistical artifacts, prompting our focus for the GPP autocorrelation on one-month lags as ecologically relevant connections. The Wetness influence on GPP demonstrated through a statistically significant relationship at zero lag across all catchments, confirms the critical role of water availability in vegetation productivity. A significant causal link strength appeared
even with a one-month lag in approximately 280 catchments. Our analysis also unveils the subtle interaction between climate and photosynthesis through the causal links between VPD and GPP. Even though the cross-correlation analysis shows VPD lagging behind GPP, the significant influence of VPD on GPP extends back up to 2 months. A larger number of catchments (282) demonstrated significant causal links at a one-month lag (i.e., GPP lagging VPD). Notably, in the VPD-GPP causal link, catchments with negative MCIs mostly appear in arid regions, reflecting the plants' defensive mechanism of stomatal closure
under high VPD conditions, leading to reduced carbon uptake. Conversely, a positive MCI in humid climates validates the stimulating impact of moderate VPD levels on GPP through enhanced transpiration. As lags increased, a decline in the number of catchments and the strength of significant causal links suggest a diminishing effect over time, stressing the importance of considering temporal lags in ecological modeling. These observations provide a crucial foundation for future research and could guide the development of more accurate and region-specific eco-hydrological models. Overall, the causal analysis sup-
ports the lag correlation analysis, indicating that most of the cause-effect relationship between Wetness and GPP, and VPD and GPP spans a maximum of two months.

## 6 Conclusions

This study employs a comparative analysis to investigate the lag in vegetation productivity response to catchment wetness and atmospheric dryness, utilizing 341 catchments distributed across topographic, climatic, and vegetation gradients of the
385 contiguous US. Using comparative analysis, we investigated the intra-annual variability and connectedness between catchment water available for vegetation use, atmospheric water demand, and vegetation carbon uptake. Our primary objective was to evaluate the interactions between these variables, particularly the controlling factors at the catchment scale. These controlling factors could provide insights into the causal relationships between the variables. However, the questions that emerged from our findings remain: How robust are these causal links? Furthermore, we aimed to determine if specific periods are critical
drivers for these links. Specifically, are certain months primarily influencing the GPP−Wetness and GPP−VPD causal link?

Our correlation analysis showed a strong, inverse relationship between Wetness and VPD on an annual scale. Yet, this pattern seemed to stem from a few highly productive months predominantly. Correlation analysis of GPP with Wetness and



VPD at a monthly scale revealed a stronger connection during these specific months across all catchments. Further, our cross-correlation analysis showed a lag in the cause-effect relationships between water supply, atmospheric demand, and vegetation productivity from 0 to 2 months. Moreover, Granger causality tests also support that Wetness and VPD have a statistically significant impact on GPP across all catchments, emphasizing the compound effect of these factors on the seasonal dynamics of catchment GPP. The study further explored the spatial relationship between GPP, Wetness, and VPD across different vegetation classes, revealing various interactions across vegetation types and seasons. Most notably, strong positive correlations between GPP and Wetness were observed in catchments in water-limited vegetation types during non-growing periods, whereas a negative correlation emerged during the peak growing season. These findings imply a swift response of vegetation to changes in water supply during non-growing periods but a delayed response during peak productivity months, highlighting the temporal sensitivity of vegetation to changes in water availability.

Vegetation response lagged behind changes in Wetness, and changes in VPD followed the vegetation response, resulting in a hysteresis phenomenon. The sizes of this hysteresis varied, reflecting diverse vegetation responses to shifts in Wetness and VPD across various catchments and vegetation types. We conducted PCA using selected variables that had a significant correlation with the areas of the GPP−Wetness and GPP−VPD hysteresis loops. The analysis showed that the first two principal components accounted for more than 80% of the variability of the size of the hysteresis loops across catchments. This finding points to long-term properties as fundamental drivers of the differences between catchments. It is also worth noting that other sets of long-term catchment properties could potentially explain this variability to a similar extent.

Our causality analysis revealed a strong positive causal link between the current and the preceding month's GPP, reflecting the temporal continuity typical of ecological processes. We also found a significant positive causal link between Wetness and GPP with no lag and at a one-month lag. The VPD-GPP relationship exhibited a significant link with a delay of up to two months, with a positive connection in humid climates and a negative one in arid regions. Collectively, these causality analysis results indicate that the cause-effect relationship between catchment water supply and productivity, as well as atmospheric demand and GPP, spans a maximum period of two months. These findings offer valuable insights into the mechanisms and patterns of vegetation responses to changes in water availability, underlining the need to account for these factors in vegetation productivity models.

*Data availability.* Catchment hydrometeorological data used in this study come from the CAMELS dataset (https://ral.ucar.edu/solutions/products/camels). Data for GPP are sourced from the Continental US Landsat product provided by the University of Montana Numerical Terradynamic Simulation Group (NTSG) through Google Earth Engine (https://developers.google.com/earth-engine/datasets/catalog/UMT_NTSG_v2_LANDSAT_GPP). The Leaf Area Index data is obtained from AVHRR, also available through Google Earth Engine (https://developers.google.com/earth-engine/datasets/catalog/NOAA_CDR_AVHRR_LAI_FAPAR_V5).



*Author contributions.* GWA and HYL jointly conceived the idea and designed the study. GWA performed the data analysis and prepared the figures. All authors contributed to the discussion and manuscript writing.

*Competing interests.* The authors declare no conflicts of interest in regard to this manuscript.

*Acknowledgements.* This research was supported as part of the Energy Exascale Earth System Model (E3SM) project, funded by the US Department of Energy, Office of Science, Office of Biological and Environmental Research as part of the Earth System Model Development program area. M. Shi was partly supported by the US Department of Energy Office of Science Biological and Environmental Research as part of the Terrestrial Ecosystem Science Program through the Next-Generation Ecosystem Experiments (NGEE) Tropics project. The Pacific
Northwest National Laboratory is operated by Battelle for the US Department of Energy under Contract DE-AC05-76RLO1830.



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







**Figure 1.** (a) Spatial distribution of the 341 study catchments across the contiguous United States, with catchments color-coded based on their long-term mean annual green vegetation fraction. (b) Scatterplot showing the relationship between the mean annual ecological aridity index (PET / W) and the mean annual Horton index (ET / W) for the study catchments. The dashed line represents the energy limit (ET=PET) and water limit (ET = P). (c) Number of catchments within each dominant vegetation type: Evergreen Forest (EF), Deciduous Broadleaf Forest (DBF), Mixed Forest (MF), Woody Savannas and Shrublands (WSSL), Grasslands (GL), and Croplands/Natural Vegetation Mosaic (CL/NVM).





**Figure 2.** Heatmaps showing monthly Pearson's r indicating within catchment relationships between a) Wetness-VPD, b) GPP-VPD, and c) GPP-Wetness. The monthly Pearson correlations for each catchment are computed independently. Vegetation types include Evergreen Forest (EF), Deciduous Broadleaf Forest (DBF), Mixed Forest (MF), Woody Savannas and Shrublands (WSSL), Grasslands (GL), and Croplands/Natural Vegetation Mosaic (CL/NVM).





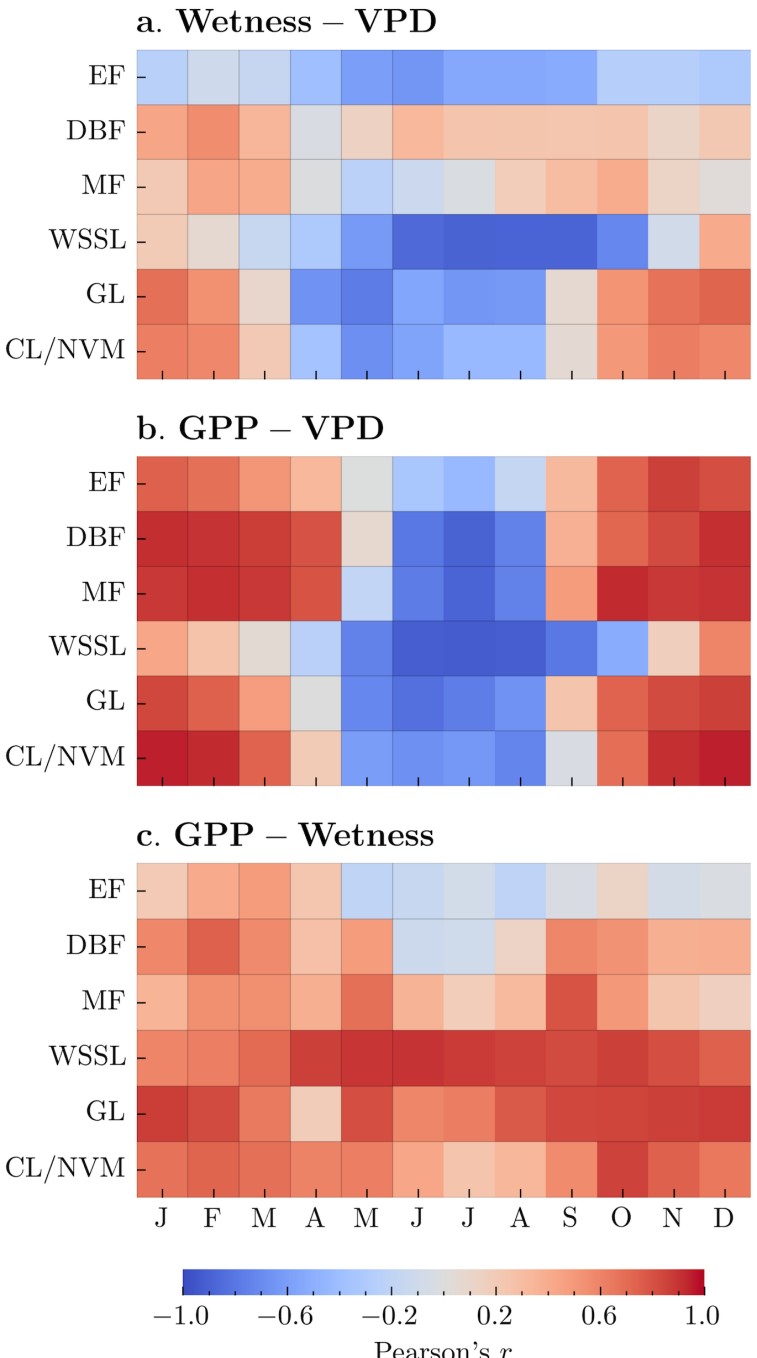

**Figure 3.** Circularity statistics for (a) VPD, (b)Wetness and (c) GPP. The SI values range from 0 to 1, with higher values indicating stronger seasonality. The orientation of the arrows indicates the average time of occurrence $\varphi$, which should be judged relative to the provided four main directions.





**Figure 4.** Hysteresis patterns between normalized GPP and Wetness for the six vegetation groups. The variables are normalized by their mean values exceeding the 90th percentile. The dashed line represents the median hysteresis curve. The letters on the color bar represent months, with J for January, F for February, and so on.



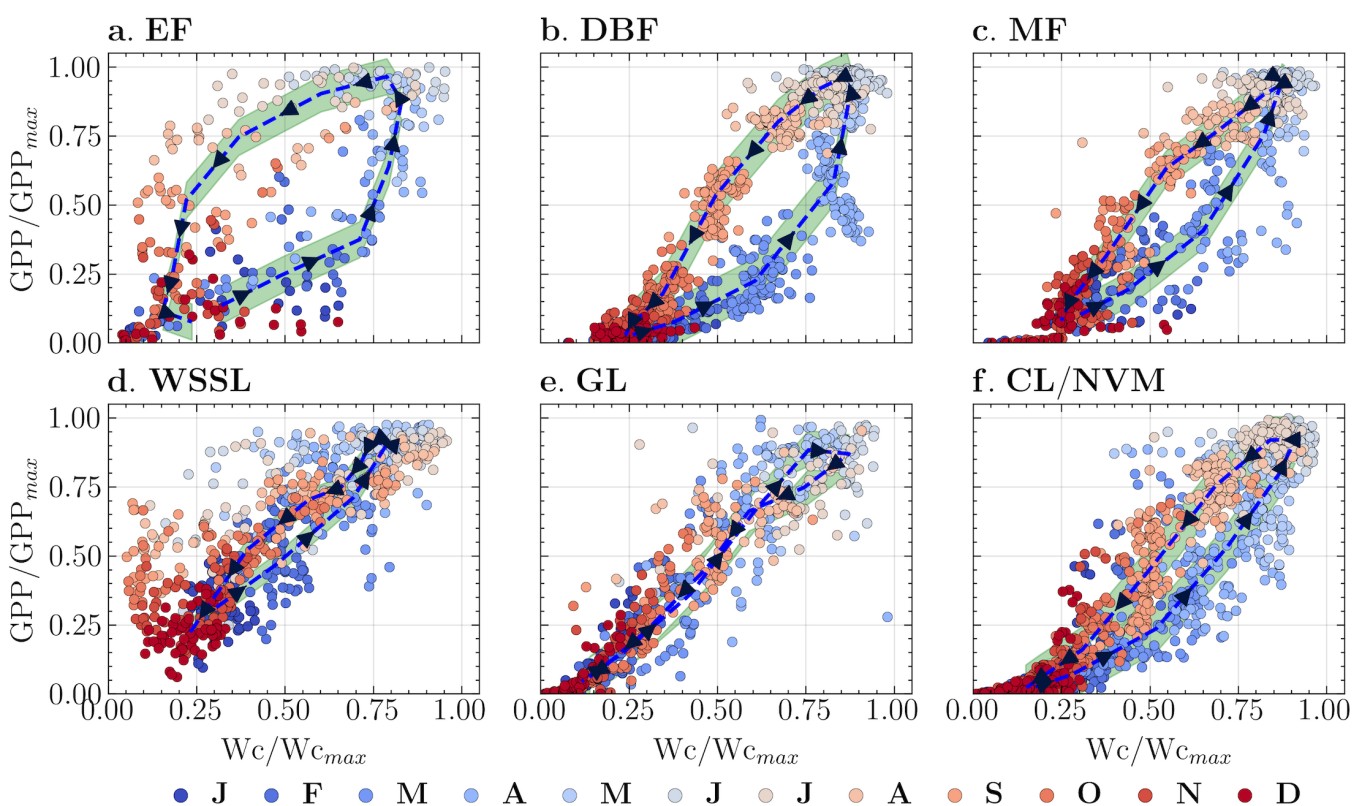

**Figure 5.** Heatmaps showing the Pearson's r indicating between catchments relationships for (a) Wetness and VPD, (b) GPP and VPD, and (c) GPP and Wetness for each vegetation type. The color scale represents the strength and direction of the correlations, with blue indicating negative correlations and red indicating positive correlations.



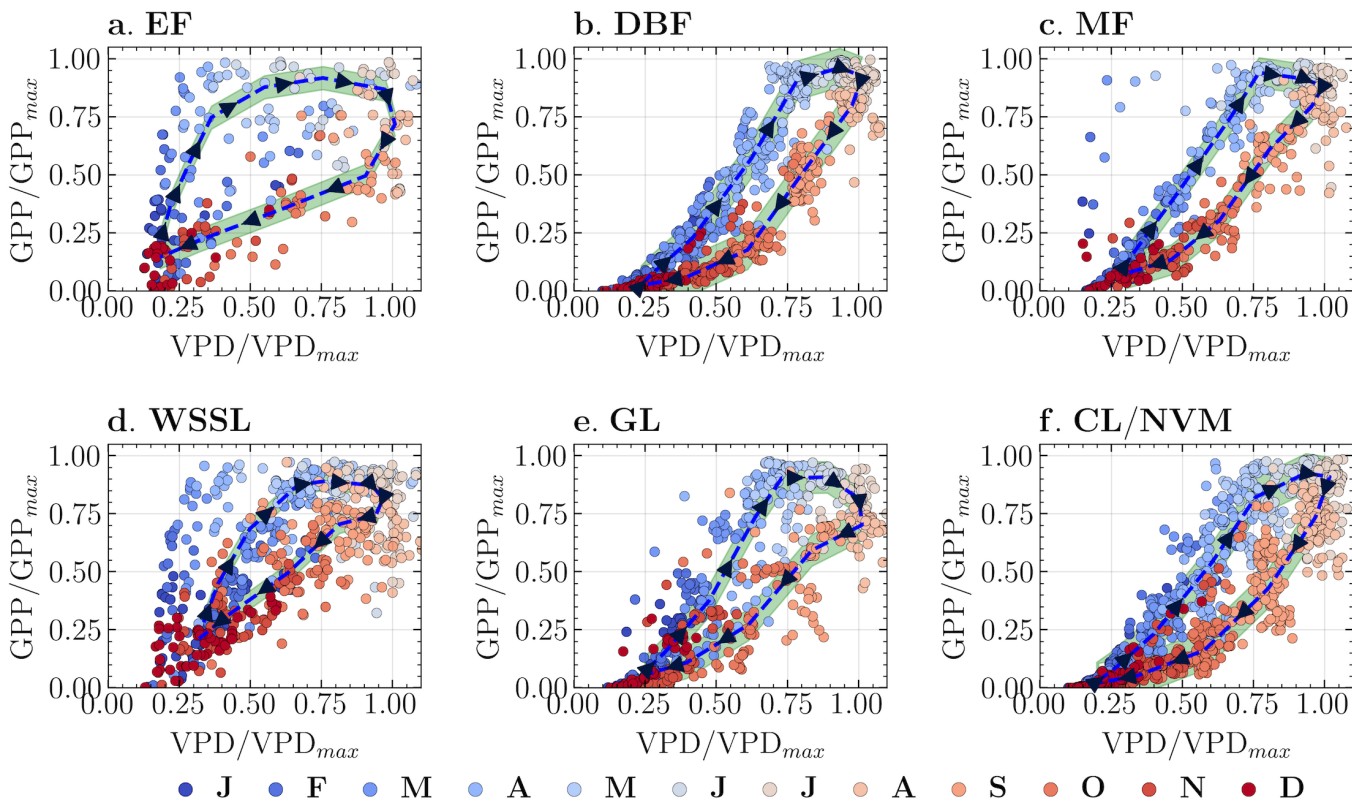

**Figure 6.** Hysteresis patterns between normalized GPP and VPD for the six vegetation groups. The dashed line represents the median hysteresis curve. The letters on the color bar represent months, with J for January, F for February, and so on.





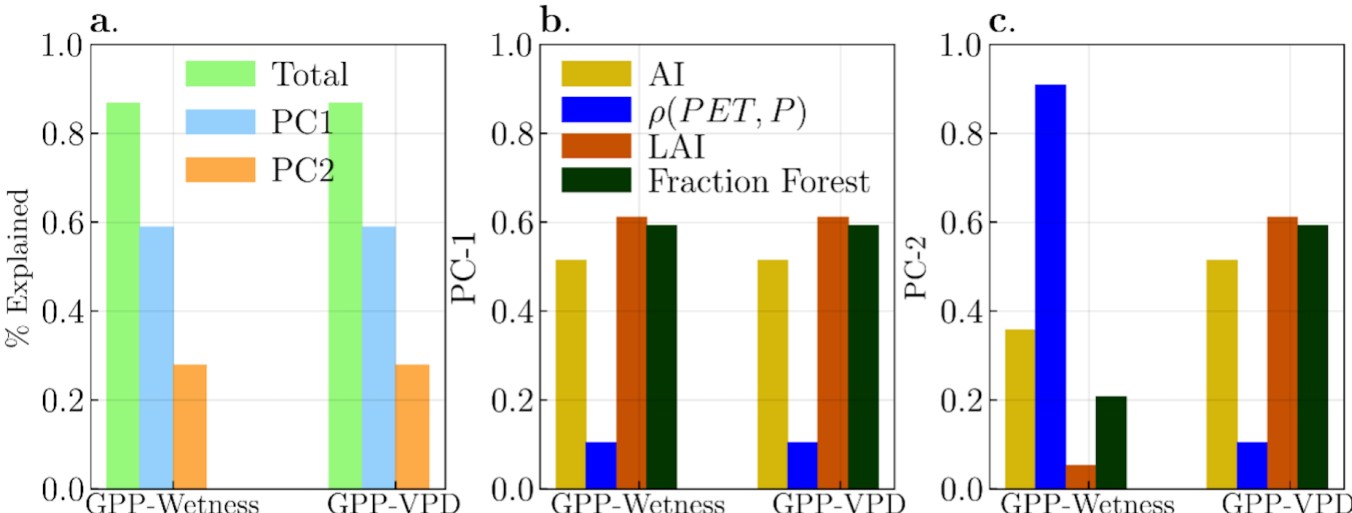

**Figure 7.** Principal Component Analysis (PCA) results showing the relationships between the hysteresis loop area and long-term catchment characteristics for (a) GPP-Wetness and GPP-VPD hysteresis. (a) The first two principal components (PC-1 and PC-2) collectively account for more than 80% of the variability in the loop sizes for both GPP-Wetness and GPP-VPD hysteresis. (b) and (c) show the contributions of the identified variables to the variability along PC-1 and PC-2 for GPP-Wetness and GPP-VPD hysteresis, respectively.



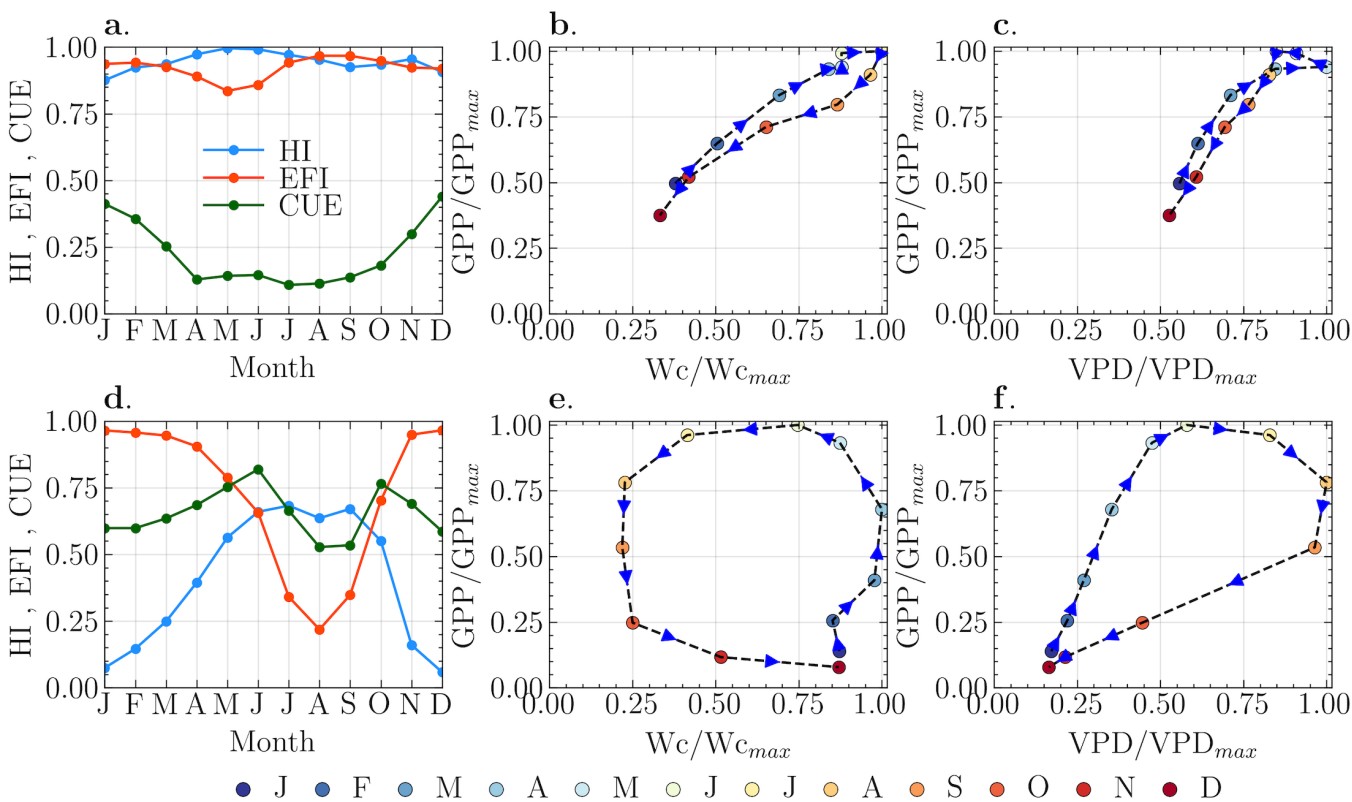

**Figure 8.** GPP-Wetness and GPP-VPD hysteresis patterns. Upper row is for narrow hysteresis and bottom row is for wide hysteresis. The arrows on b, c, e and f indicate the direction of hysteresis.







**Figure 9.** Spatial patterns of the causal link strength, measured by the Momentary Conditional Independence (MCI), monthly GPP with its past months values and (b, e, h, k), GPP and Wetness (c, f, i, l) and GPP and VPD (b, e, h, k) at 0, 1, 2, and 3 months lag. The color scale indicates the MCI values, ranging from negative (blue) to positive (red), reflecting the strength and direction of the causal link.