# Peer review of "Causal relationships between vegetation productivity, water availability, and atmospheric dryness at the catchment scale"

_EGUsphere, 2024_

## Author Comment (AC1)

**Reviewer # RC1**

The authors investigated the causal relationships between catchment water availability, vapor pressure deficit, and gross primary productivity across 341 catchments in the contiguous US. They employed various statistical methods, including circularity statistics, correlation analysis, and causality tests, to determine the complex interactions between catchment wetness, atmospheric dryness, and vegetation carbon uptake. I found this work interesting, as it enhances our understanding and predictive capabilities regarding ecosystem responses to climate change. I have few minor suggestions:

*We would like to express our sincere gratitude for your positive feedback regarding our manuscript. We are pleased that you found our work interesting and valuable for enhancing the understanding of ecosystem responses to climate change. Your insightful comments and suggestions are greatly appreciated, and we have carefully considered each one.*

Comment: What are the potential reasons for a strong positive causal link between the current and the preceding month's GPP?

Response*: We appreciate your insightful question. This relationship is indeed complex and multifaceted, stemming from several interconnected ecological and physiological factors. We provide a couple of examples: 1) Biological Inertia: vegetation exhibits a form of biological inertia, where its physiological state in one month significantly influences productivity in the following month. For instance, the continuity in the leaf area index reflects gradual changes in canopy structure, leading to a strong month-to-month correlation in photosynthetic capacity. Additionally, the development of the root system over time affects water and nutrient uptake in subsequent periods. 2) Soil Moisture Memory: Soil water content often has a memory effect that can span weeks to months, impacting plant water availability and, consequently, GPP.*

Comment: This is an interesting finding: "Vegetation response lagged behind changes in Wetness, and changes in VPD followed the vegetation response, resulting in a hysteresis phenomenon." Please comment if such hysteresis phenomena are likely to change in space and time.

Response:  *Thank you for highlighting this important aspect of our findings. We address the potential changes in hysteresis phenomena in both spatial and temporal contexts. Spatial Changes: Across CONUS watersheds, we observed that the size and nature of the hysteresis phenomenon vary geographically. This variation is driven by differences in factors such as seasonal dynamics of hydrologic dryness and vegetation carbon uptake efficiency. We will revise the manuscript to provide greater clarity and detail on this spatial variability, ensuring that the geographical differences in hysteresis are more explicitly articulated. Temporal Changes: Our study utilized regime curves based on long-term monthly averages, which limits our ability to evaluate temporal changes in hysteresis within the study period. To assess temporal variability, it would be necessary to divide the study period into smaller segments and analyze them individually. We plan to address this task in future research, where we will explore temporal shifts in hysteresis patterns over shorter time scales. We appreciate your suggestion and will incorporate additional explanations in the revised manuscript to enhance the discussion of these spatial and temporal dynamics. This*

*will help clarify the potential for hysteresis phenomena to change over space and time, providing a more comprehensive understanding of these processes.*

Comment: Cations for figures 3-5 seem to be swapped. Please correct.
Response: *Thank you for bringing this to our attention. We will review and correct the captions for Figures 3-5 in the revised manuscript.*

Comment: In Section 5, please develop your discussion in the context of prior similar studies and articulate your major contributions.
Response: *Thank you for your suggestion. We will revise Section 5 to better contextualize our findings within prior studies and clearly articulate our major contributions.*

Comment: In Section 5 or 6, please include one paragraph on the limitations of your study. It is often quite complex to study the non-linear relationship between selected variables.
Response: *Thank you for this valuable suggestion. We will provide more discussion on the limitations of our study is crucial for providing a balanced perspective on our findings.*

Comment: Can we use Soil moisture products (e.g., remote sensing products) instead of W minus (deltaS)?

Response: *Thank you for this insightful question. We chose to use W minus (deltaS) instead of soil moisture products for several important reasons. W minus (deltaS) represents the total water available to vegetation, including not only soil moisture but also deeper groundwater resources, which is crucial because many plants, especially those with deep root systems, can access water beyond the shallow soil layers typically measured by remote sensing products. Satellite-based soil moisture products generally capture moisture only in the top few centimeters of soil, leading to an incomplete picture of water availability for vegetation. By using W minus (deltaS), we aimed to move beyond the traditional reliance on surface soil moisture, which can be limiting in ecosystem studies, and provide a more integrated measure of water availability at the catchment scale that aligns better with our study's spatial focus. Additionally, W minus (deltaS) is more directly linked to the overall water balance of the catchment, offering a more holistic representation of water availability. While soil moisture products have their merits for surface-level analyses, our choice of W minus (deltaS) allows for a more comprehensive assessment of water dynamics relevant to vegetation across various depths and ecosystem types. We appreciate your question as it allows us to clarify this important methodological choice in our study.*

Comment: Please justify why you made 6 groups to represent 341 catchments.
Response: *Thank you for your question. The decision to group the 341 catchments into six categories was based on the dominant vegetation cover, which is defined as covering more than 50% of the watershed area. This information is sourced from the CAMELS dataset and aligns with the National Land Cover Database (NLCD) classifications. To streamline our analysis and facilitate meaningful comparisons, we consolidated similar vegetation classes into six broader categories. Evergreen Forest: This group combines Broadleaf and Needleleaf Evergreen Forests, reflecting their similar ecological functions and carbon uptake dynamics. Woody Savannah, Open*

*and Closed Shrublands: These classes were merged due to their comparable structural and functional characteristics to form Woody Savannah and Shrublands group. Cropland/Natural Vegetation Mosaic (NVM) and Cropland: These were grouped together to account for areas dominated by agricultural activities. The other three groups including Deciduous Broadleaf Forest (DBF), Grasslands (GL) and Mixed Forest are all original classification from NLCD and has not been merged with any other group. This categorization allowed us to efficiently analyze and interpret the data across the catchments, ensuring that each group represented a distinct ecological and hydrological regime. We will include this explanation in Section 2 of the manuscript to provide clarity on our methodological approach.*

Comment: How did you analyze various data sets when they have different spatial and temporal resolutions? For example, the GPP dataset features a spatial resolution of 30 meters and a temporal resolution of 16 days, while other data sets are of varying resolution. How it was handled in the analysis?

Response: *Thank you for your question. To address the challenge of integrating datasets with different spatial and temporal resolutions, we conducted our analysis monthly with the watershed as the spatial unit. For climate variables, we used the CAMELS dataset, which provides spatially averaged daily data for each watershed. This daily data was aggregated to a monthly scale to align with our analysis timeframe. Similarly, the daily streamflow data can be converted to daily runoff depth and then to monthly scale. For the GPP data, which is available at a 16-day temporal resolution, we first converted the values to a daily scale by assuming a uniform distribution of GPP over each 16-day period. We then aggregated these daily values to a monthly scale to ensure consistency with other datasets. Spatially, we clipped the GPP data, originally at a 30-meter resolution, to match the watershed polygons and calculated the spatial average of all 30-meter pixels within each polygon. This approach allowed us to derive a representative GPP value for each watershed, maintaining spatial consistency with other datasets. By standardizing both the temporal and spatial scales in this manner, we ensured that all datasets were compatible for integrated analysis. We will include this explanation in the methodology section of the manuscript to clarify our data processing approach.*

---

## Author Comment (AC2)

**Reviewer #RC2**

This paper investigated the relationships among catchment water availability, vapor pressure deficit, and gross primary productivity using causality analysis, circularity statistics, Principal Component Analysis, etc. The topic is novel and meaningful, the findings are interesting. Here are some concerns and suggestions:

*Thank you for your positive feedback on our manuscript. We appreciate your recognition of the novelty and significance of our work, as well as your interest in our findings. We are committed to addressing your concerns and suggestions in detail and believe they will enhance the quality of our paper. We look forward to incorporating your insights in our revision.*

Comment: Lines 95 to 99: How did you divide the catchments into six vegetation groups? Find the primary vegetation type based of the percentage of each vegetation type in the catchment? What are the criteria?

Response: *Thank you for your question. The decision to group the 341 catchments into six categories was based on the dominant vegetation cover, which is defined as covering more than 50% of the watershed area. This information is sourced from the CAMELS dataset and aligns with the National Land Cover Database (NLCD) classifications. To streamline our analysis and facilitate meaningful comparisons, we consolidated similar vegetation classes into six broader categories. Evergreen Forest: This group combines Broadleaf and Needleleaf Evergreen Forests, reflecting their similar ecological functions and carbon uptake dynamics. Woody Savannah, Open and Closed Shrublands: These classes were merged due to their comparable structural and functional characteristics to form the Woody Savannah and Shrublands group. Cropland/Natural Vegetation Mosaic (NVM) and Cropland: These were grouped together to account for areas dominated by agricultural activities. The other three groups, including Deciduous Broadleaf Forest (DBF), Grasslands (GL), and Mixed Forest, are all original classifications from NLCD and have not been merged with any other group. This categorization allowed us to efficiently analyze and interpret the data across the catchments, ensuring that each group represented a distinct ecological and hydrological regime. We will include this explanation in Section 2 of the manuscript to provide clarity on our methodological approach.*

Comment: Lines 218 to 224, what are the reasons causing the different lag time (e.g., 0, 1 month, 2 month) from the perspective of catchment chrematistics?

Response: *Thank you for this insightful question. The different lag times between water availability and GPP, as well as between GPP and VPD, are influenced by catchment characteristics such as soil moisture dynamics, vegetation response, and hydrology. Faster water infiltration in well-drained soils results in shorter lags between water availability and GPP, whereas slower water release in clay-rich soils extends these lags. Dense or deep-rooted vegetation may delay the response to changes in water availability, leading to a longer lag between GPP and VPD as transpiration continues after peak GPP. Additionally, catchments with significant groundwater contributions or unique climate patterns may experience extended lags due to sustained water availability. We will include a discussion of these factors in the revised manuscript to clarify the observed variations in lag times.*

**Comment:** You used PCA and found that the first two principal components accounted for most of the variability of the size of the hysteresis loops across catchments. Did you also research on the importance of those selected variables used for PCA to see the dominant factors?

**Response:** *Thank you for the question. While we focused on using PCA to capture the main sources of variability in the size of the hysteresis loops, we did not conduct a separate analysis to evaluate the importance of the individual variables used in the PCA. However, this is an excellent suggestion, and we plan to explore the dominant factors in future work. We will also consider including a brief discussion of potential dominant variables in the revised manuscript to provide additional context.*

**Comment:** The last two paragraphs in the Discussion section seem more like results and conclusions. I suggest adding a discussion about whether any previous studies support or contradict your findings.

**Response:** *Thank you for your observation. We will revise the stated paragraphs and include a discussion on how our findings align with or differ from previous studies.*

**Comment:** Lines 119 – 120: The sentence has grammar error.

**Response:** *Thank you. This will be corrected in the revised manuscript*

**Comment:** For Figure 4, the title states that "The letters on the color bar represent months, with J for January, F for February, and so on.", please double check it.

**Response:** *Thank you for pointing this out. Somehow this is shifted up from Figure 5. This will be corrected in the revised manuscript*

---

## Author Response (AR1)

**Reviewer # RC1**

The authors investigated the causal relationships between catchment water availability, vapor pressure deficit, and gross primary productivity across 341 catchments in the contiguous US. They employed various statistical methods, including circularity statistics, correlation analysis, and causality tests, to determine the complex interactions between catchment wetness, atmospheric dryness, and vegetation carbon uptake. I found this work interesting, as it enhances our understanding and predictive capabilities regarding ecosystem responses to climate change. I have few minor suggestions:

*We would like to express our sincere gratitude for your positive feedback regarding our manuscript. We are pleased to hear that you found our work interesting and valuable for enhancing the understanding of ecosystem responses to climate change. Your insightful comments and suggestions are greatly appreciated, and we have carefully considered each one.*

Comment: What are the potential reasons for a strong positive causal link between the current and the preceding month's GPP?

Response*: We appreciate your insightful question. This relationship is indeed complex and multifaceted, stemming from several interconnected ecological and physiological factors, we provide a couple of examples. 1) Biological Inertia: vegetation exhibits a form of biological inertia, where its physiological state in one month significantly influences productivity in the following month. For instance, the continuity in leaf area index reflects gradual changes in canopy structure, leading to a strong month-to-month correlation in photosynthetic capacity. Additionally, the development of the root system over time affects water and nutrient uptake in subsequent periods. 2) Soil Moisture Memory: Soil water content often has a memory effect that can span weeks to months, impacting plant water availability and, consequently, GPP. We have added the following sentences on line 445.*

"The strong causal link between the current and preceding month's GPP can be attributed to vegetation's biological inertia (where a plant's physiological state influences future productivity), soil moisture memory effects (as soil water content impacts plant water availability over time), and consistent phenological patterns (reflecting seasonal growth trends), all of which contribute to the continuity of productivity across consecutive months, among other things."

Comment: This is an interesting finding: "Vegetation response lagged behind changes in Wetness, and changes in VPD followed the vegetation response, resulting in a hysteresis phenomenon." Please comment if such hysteresis phenomena are likely to change in space and time.

Response: *Thank you for highlighting this important aspect of our findings. We address the potential changes in hysteresis phenomena in both spatial and temporal contexts.*
*Spatial Changes: across CONUS watersheds, we observed that the size and nature of the hysteresis phenomenon vary geographically. This variation is driven by differences we have investigated and discussed it (see Fig. 7), factors such as aridity, the sync between seasonal patterns of PET and P, LAI, and the fraction of the catchment covered by forest found to be*

*important factors in defining the size of the hysteresis, hence the spatial differences between catchments. Temporal Changes: Our study utilized regime curves based on long-term monthly averages, which limits our ability to evaluate temporal changes in hysteresis within the study period. To assess temporal variability, it would be necessary to divide the study period into smaller segments and analyze them individually. This is a task we plan to address in future research, where we will explore temporal shifts in hysteresis patterns over shorter time scales. We appreciate your suggestion; we hope this will help clarify the potential for hysteresis phenomena to change over space and time.*

Comment: Cations for figures 3-5 seem to be swapped. Please correct.
Response: *Thank you for bringing this to our attention. The captions are now corrected in the revised manuscript.*

Comment: In Section 5, please develop your discussion in the context of prior similar studies and articulate your major contributions.
Response: *Thank you for your suggestion. The following paragraph is added to section 5.*

"This study builds upon the foundational understanding of interactions between soil moisture, VPD, and vegetation productivity highlighted in previous research. For instance, Liu et al. (2020) identified soil moisture as a key driver of ecosystem productivity in over 70% of vegetated land areas globally. We extend this understanding by evidencing significant positive causal links between Wetness and GPP across 341 diverse catchments in the contiguous United States. Similarly, our identification of strong negative correlations between VPD and GPP during peak growing seasons corroborates the findings of Novick et al. (2016) and Giardina et al. (2018), both of whom underscored the substantial role of VPD in modulating vegetation carbon-water exchange. Moreover, our analysis of hysteresis patterns in GPP-Wetness and GPP-VPD relationships provides novel insights into lagged vegetation responses to hydrological and atmospheric variables. While Zhou et al. (2014) observed such lag characteristics at diurnal scales, our study extends these observations to longer temporal scales and diverse catchment types. Building on Zhou et al. (2019), who found that reduced soil moisture can lead to extreme atmospheric aridity through feedback mechanisms, our work further emphasizes the intricate relationships between soil moisture, VPD, and vegetation productivity. Our comprehensive analysis across various ecosystems and climatic regimes enhances the spatial and temporal nuance of these relationships. Notably, the application of circularity statistics and hysteresis analysis reveals new temporal dynamics, particularly the lag times between changes in water availability, atmospheric demand, and vegetation response. This methodological approach builds on the diurnal scale observations of Zhou et al. (2014), extending them to longer time scales and across diverse catchments. Furthermore, our causality analysis, employing Granger causality tests and PCMCI+, provides a quantitative basis for determining the direction and strength of causal links between these variables."

Comment: In Section 5 or 6, please include one paragraph on the limitations of your study. It is often quite complex to study the non-linear relationship between selected variables.
Response: *Thank you for this valuable suggestion. The following paragraph is now added to section 5*

"While our study provides valuable insights into the complex interactions between vegetation productivity, water availability, and atmospheric dryness, it is important to acknowledge certain limitations. The non-linear relationships between GPP, Wetness, and VPD present significant challenges in their analysis and interpretation. Our approach, while comprehensive, may not fully capture all the nuances of these complex interactions. For instance, the use of monthly aggregated data may obscure finer-scale temporal dynamics that could be important in understanding rapid ecosystem responses to changes in water availability or atmospheric demand. Additionally, while our causality analysis provides important insights, it is based on statistical relationships and may not always reflect true mechanistic causality. The spatial heterogeneity within catchments, which can influence local water availability and vegetation responses, is not fully accounted for in our catchment-scale analysis. Furthermore, our study does not explicitly consider the effects of extreme events or long-term climate change, which could potentially alter the relationships we've observed. Future studies could address these limitations by incorporating higher temporal resolution data, considering spatial heterogeneity within catchments, and explicitly modeling non-linear relationships and feedback mechanisms between variables."

Comment: Can we use Soil moisture products (e.g., remote sensing products) instead of W minus (deltaS)?

Response: *Thank you for this insightful question. We chose to use W-ΔS instead of soil moisture products for several important reasons. W-ΔS represents the total water available to vegetation, including not only soil moisture but also deeper groundwater resources, which is crucial because many plants, especially those with deep root systems, can access water beyond the shallow soil layers typically measured by remote sensing products. Satellite-based soil moisture products generally capture moisture only in the top few centimeters of soil, leading to an incomplete picture of water availability for vegetation. By using W-ΔS, we aimed to move beyond the traditional reliance on surface soil moisture, which can be limiting in ecosystem studies, and provide a more integrated measure of water availability at the catchment scale that aligns better with our study's spatial focus. Additionally, W-ΔS) is more directly linked to the overall water balance of the catchment, offering a more holistic representation of water availability. While soil moisture products have their merits for surface-level analyses, our choice of W-ΔS allows for a more comprehensive assessment of water dynamics relevant to vegetation across various depths and ecosystem types. That said, a remote sensing product that could be equally useful to W-ΔS could be the total water storage from the GRACE satellite. However, this data cannot be used for catchment-level study because of its coarse spatial resolution. We appreciate your question; we hope this clarifies our preference.*

Comment: Please justify why you made 6 groups to represent 341 catchments.
Response: *Thank you for your question. The decision to group the 341 catchments into six categories was based on the dominant vegetation cover, which is defined as covering more than 50% of the watershed area. This information is sourced from the CAMELS dataset and aligns with the National Land Cover Database (NLCD) classifications. To streamline our analysis and facilitate meaningful comparisons, we consolidated similar vegetation classes into six broader categories. Evergreen Forest: This group combines Broadleaf and Needleleaf Evergreen Forests, reflecting their similar ecological functions and carbon uptake dynamics. Woody Savannah, Open and Closed Shrublands: These classes were merged due to their comparable structural and*

*functional characteristics to form the Woody Savannah and Shrublands group. Cropland/Natural Vegetation Mosaic (NVM) and Cropland: These were grouped together to account for areas dominated by agricultural activities. The other three groups, including Deciduous Broadleaf Forest (DBF), Grasslands (GL), and Mixed Forest, are all original classifications from NLCD and have not been merged with any other group. This categorization allowed us to efficiently analyze and interpret the data across the catchments, ensuring that each group represented a distinct ecological and hydrological regime. The following clarifying paragraph is added at the end of Section 2.*

"These 341 catchments are divided into six vegetation groups based on the dominant vegetation cover vegetation covering 50% of the catchments area). The dominant vegetation cover information is sourced from the CAMELS dataset and aligns with the National Land Cover Database (NLCD) classifications. To streamline our analysis and facilitate meaningful comparisons, we consolidated similar vegetation classes into six broader categories. Evergreen Forest (EF): this group combines Broadleaf and Needleleaf Evergreen Forests, reflecting their similar ecological functions and carbon uptake dynamics. Woody Savannah, Open and Closed Shrublands: these classes were merged due to their comparable structural and functional characteristics to form Woody Savannah and Shrublands (WSSL) group. Cropland/Natural Vegetation Mosaic (CL/NVM) and Cropland (CL) were grouped together as CL/NVM to account for areas dominated by agricultural activities. The other three groups including Deciduous Broadleaf Forest (DBF), Grasslands (GL) and Mixed Forest (MF) are all original classification from NLCD and has not been merged with any other group. This categorization allowed us to efficiently analyze and interpret the data across the catchments, ensuring that each group represented a distinct dominant ecological regime. Among the 341 catchments, we have 101 CL/NVM, 85 DBF, 51 WSSL, 43 GL, 40 MF and 21 EF catchments"

Comment: How did you analyze various data sets when they have different spatial and temporal resolutions? For example, the GPP dataset features a spatial resolution of 30 meters and a temporal resolution of 16 days, while other data sets are of varying resolution. How it was handled in the analysis?

Response: *Thank you for your question. To address the challenge of integrating datasets with different spatial and temporal resolutions, we conducted our analysis monthly with the watershed as the spatial unit. For climate variables, we used the CAMELS dataset, which provides spatially averaged daily data for each watershed. This daily data was aggregated to a monthly scale to align with our analysis timeframe. Similarly, the daily streamflow data can be converted to daily runoff depth and then to monthly scale. For the GPP data, which is available at a 16-day temporal resolution, we first converted the values to a daily scale by assuming a uniform distribution of GPP over each 16-day period. We then aggregated these daily values to a monthly scale to ensure consistency with other datasets. Spatially, we clipped the GPP data, originally at a 30-meter resolution, to match the watershed polygons and calculated the spatial average of all 30-meter pixels within each polygon. This approach allowed us to derive a representative GPP value for each watershed, maintaining spatial consistency with other datasets. By standardizing both the temporal and spatial scales in this manner, we ensured that all datasets were compatible for integrated analysis. The explanation can be found in lines 85 - 90.*

**Reviewer #RC2**

This paper investigated the relationships among catchment water availability, vapor pressure deficit, and gross primary productivity using causality analysis, circularity statistics, Principal Component Analysis, etc. The topic is novel and meaningful, the findings are interesting. Here are some concerns and suggestions:

*Thank you for your positive feedback on our manuscript. We appreciate your recognition of the novelty and significance of our work, as well as your interest in our findings. We are committed to addressing your concerns and suggestions in detail and believe they will enhance the quality of our paper. We look forward to incorporating your insights in our revision.*

Comment: Lines 95 to 99: How did you divide the catchments into six vegetation groups? Find the primary vegetation type based of the percentage of each vegetation type in the catchment? What are the criteria?

Response: *Thank you for your question. The decision to group the 341 catchments into six categories was based on the dominant vegetation cover, which is defined as covering more than 50% of the watershed area. This information is sourced from the CAMELS dataset and aligns with the National Land Cover Database (NLCD) classifications. To streamline our analysis and facilitate meaningful comparisons, we consolidated similar vegetation classes into six broader categories. Evergreen Forest: This group combines Broadleaf and Needleleaf Evergreen Forests, reflecting their similar ecological functions and carbon uptake dynamics. Woody Savannah, Open and Closed Shrublands: These classes were merged due to their comparable structural and functional characteristics to form the Woody Savannah and Shrublands group. Cropland/Natural Vegetation Mosaic (NVM) and Cropland: These were grouped together to account for areas dominated by agricultural activities. The other three groups, including Deciduous Broadleaf Forest (DBF), Grasslands (GL), and Mixed Forest, are all original classifications from NLCD and have not been merged with any other group. This categorization allowed us to efficiently analyze and interpret the data across the catchments, ensuring that each group represented a distinct ecological and hydrological regime. The following clarifying paragraph is added at the end of Section 2.*

"These 341 catchments are divided into six vegetation groups based on the dominant vegetation cover vegetation covering 50% of the catchments area). The dominant vegetation cover information is sourced from the CAMELS dataset and aligns with the National Land Cover Database (NLCD) classifications. To streamline our analysis and facilitate meaningful comparisons, we consolidated similar vegetation classes into six broader categories. Evergreen Forest (EF): this group combines Broadleaf and Needleleaf Evergreen Forests, reflecting their similar ecological functions and carbon uptake dynamics. Woody Savannah, Open and Closed Shrublands: these classes were merged due to their comparable structural and functional characteristics to form Woody Savannah and Shrublands (WSSL) group. Cropland/Natural Vegetation Mosaic (CL/NVM) and Cropland (CL) were grouped together as CL/NVM to account for areas dominated by agricultural activities. The other three groups including Deciduous Broadleaf Forest (DBF), Grasslands (GL) and Mixed Forest (MF) are all original classification from NLCD and has not been merged with any other group. This categorization allowed us to efficiently analyze and interpret the data across the catchments, ensuring that each group

represented a distinct dominant ecological regime. Among the 341 catchments, we have 101 CL/NVM, 85 DBF, 51 WSSL, 43 GL, 40 MF and 21 EF catchments"

Comment: Lines 218 to 224, what are the reasons causing the different lag time (e.g., 0, 1 month, 2 month) from the perspective of catchment chrematistics?
Response: *Thank you for this insightful question. The different lag times between water availability and GPP, as well as between GPP and VPD, are influenced by catchment characteristics such as soil moisture dynamics, vegetation response, and hydrology. Faster water infiltration in well-drained soils results in shorter lags between water availability and GPP, whereas slower water release in clay-rich soils extends these lags. Dense or deep-rooted vegetation may delay the response to changes in water availability, leading to a longer lag between GPP and VPD as transpiration continues after peak GPP. Additionally, catchments with significant groundwater contributions or unique climate patterns may experience extended lags due to sustained water availability. The following sentences are added on line 455.*

"The strong causal link between the current and preceding month's GPP can be attributed to vegetation's biological inertia (where a plant's physiological state influences future. productivity), soil moisture memory effects (as soil water content impacts plant water availability over time), and consistent phenological patterns (reflecting seasonal growth trends), all of which contribute to the continuity of productivity across consecutive months, among other things."

**Comment:** You used PCA and found that the first two principal components accounted for most of the variability of the size of the hysteresis loops across catchments. Did you also research on the importance of those selected variables used for PCA to see the dominant factors?
**Response:** *Thank you for the question. While we focused on using PCA to capture the main sources of variability in the size of the hysteresis loops, and the relative contribution of the variables considered to PC-1 and PC-2, we did not conduct a separate analysis to evaluate the importance of the individual variables. However, this is an excellent suggestion, and we plan to explore the dominant factors in future work.*

**Comment:** The last two paragraphs in the Discussion section seem more like results and conclusions. I suggest adding a discussion about whether any previous studies support or contradict your findings.
**Response:** *Thank you for your observation. The following two paragraphs discussing relevance to other studies and the current studies' limitation are added to section 5.*

"This study builds upon the foundational understanding of interactions between soil moisture, VPD, and vegetation productivity highlighted in previous research. For instance, Liu et al. (2020) identified soil moisture as a key driver of ecosystem productivity in over 70% of vegetated land areas globally. We extend this understanding by evidencing significant positive causal links between Wetness and GPP across 341 diverse catchments in the contiguous United States. Similarly, our identification of strong negative correlations between VPD and GPP during peak growing seasons corroborates the findings of Novick et al. (2016) and Giardina et al. (2018), both of whom underscored the substantial role of VPD in modulating vegetation carbon-water exchange. Moreover, our analysis of hysteresis patterns in GPP-Wetness and GPP-VPD relationships provides novel insights into lagged vegetation responses to hydrological and

atmospheric variables. While Zhou et al. (2014) observed such lag characteristics at diurnal scales, our study extends these observations to longer temporal scales and diverse catchment types. Building on Zhou et al. (2019), who found that reduced soil moisture can lead to extreme atmospheric aridity through feedback mechanisms, our work further emphasizes the intricate relationships between soil moisture, VPD, and vegetation productivity. Our comprehensive analysis across various ecosystems and climatic regimes enhances the spatial and temporal nuance of these relationships. Notably, the application of circularity statistics and hysteresis analysis reveals new temporal dynamics, particularly the lag times between changes in water availability, atmospheric demand, and vegetation response. This methodological approach builds on the diurnal scale observations of Zhou et al. (2014), extending them to longer time scales and across diverse catchments. Furthermore, our causality analysis, employing Granger causality tests and PCMCI+, provides a quantitative basis for determining the direction and strength of causal links between these variables.

While our study provides valuable insights into the complex interactions between vegetation productivity, water availability, and atmospheric dryness, it is important to acknowledge certain limitations. The non-linear relationships between GPP, Wetness, and VPD present significant challenges in their analysis and interpretation. Our approach, while comprehensive, may not fully capture all the nuances of these complex interactions. For instance, the use of monthly aggregated data may obscure finer-scale temporal dynamics that could be important in understanding rapid ecosystem responses to changes in water availability or atmospheric demand. Additionally, while our causality analysis provides important insights, it is based on statistical relationships and may not always reflect true mechanistic causality. The spatial heterogeneity within catchments, which can influence local water availability and vegetation responses, is not fully accounted for in our catchment-scale analysis. Furthermore, our study does not explicitly consider the effects of extreme events or long-term climate change, which could potentially alter the relationships we've observed. Future studies could address these limitations by incorporating higher temporal resolution data, considering spatial heterogeneity within catchments, and explicitly modeling non-linear relationships and feedback mechanisms between variables."

**Comment:** Lines 119 – 120: The sentence has grammar error.
**Response:** *Thank you. This is corrected in the revised manuscript*

**Comment:** For Figure 4, the title states that "The letters on the color bar represent months, with J for January, F for February, and so on.", please double check it.
**Response:** *Thank you for pointing this out. Somehow this is shifted up from Figure 5. This is corrected in the revised manuscript.*

---

## Author Response (AR2)

**Referee #3**

*Response*: We would like to extend our sincere gratitude for taking the time to review our manuscript and for accepting it as is. We are pleased that the revised version meets your expectations, and we appreciate your support and positive assessment. Thank you again for your time and for contributing to the quality of our manuscript.

**Referee #4**

Soil wetness and atmospheric dryness have critical impacts on ecosystem productivity and vegetation carbon uptake, yet the relationships are complex. The authors conducted various statistical analysis to explore the complex interactions between them to derive a comprehensive understanding. A lot of analyses have been conducted rigorously to investigate the correlation and the causal links. The authors have revised the manuscript corresponding to comments from previous reviewers, I think the manuscript is in a good shape, and only have a few minor suggestions:

*Response*: We appreciate your positive assessment of our revised manuscript and your recognition of the rigorous analyses we conducted to explore the intricate relationship between soil wetness, atmospheric dryness, and vegetation carbon uptake. We also thank you for recognizing our efforts in responding to previous reviewers' comments. Your feedback has been invaluable in shaping our revisions and strengthening our manuscript. Below, we provide a more detailed, point-by-point response to your comments. The point to point changes can be found on lines 65, 82, 114, 116 and 234.

*Comment (L65)*: "What are the critical features are responsible for between-catchment differences", remove the second "are".
*Response*: Thank you for pointing this out. We have removed the extra "are" in the revised manuscript in Line 65.

*Comment (Eqn. 1)*: Remove the "+" in front of DS.
*Response*: We have removed the extraneous "+" symbol from the equation.

*Comment (L116)*: Usually it is baseflow and event flow, surface runoff and subsurface runoff. It could be confusing to use baseflow and surface runoff, as subsurface runoff isn't necessarily baseflow (e.g., subsurface stormflow). Or you can use fast flow and slow flow corresponding to the two-step partitioning.
*Response*: Thank you for this clarification. We have replaced "baseflow" and "surface runoff" with "slow runoff" and "fast runoff" to reflect the two-step partitioning in Line 116.

*Comment*: It could be helpful to add some sub-titles in the Results section, and discussion section as well if applicable.
*Response*: We appreciate this suggestion. However, we prefer to keep the Results and

Discussion as continuous sections to present a clear, uninterrupted narrative. We have structured the text in a way that introduces the key findings in a logical flow without explicit sub-headings. We hope this approach helps readers follow the arguments smoothly.

***Comment (L235):*** It helps to include the Granger causality test results used for decision, i.e., p-values.

***Response***: There is no enough space, and not necessary either, to include the p-values for all the catchments involved in this study. We included the criteria "p-values < 0.05" in Line 234 in the revised manuscript to clarify how we determined the significance of the Granger causality results.

***Comment (L265):*** I may have missed somewhere, the "occurrence" of what? Maximum Wetness and GPP? It helps to state in Section 3.2.

***Response***: Thank you for the suggestion. The "occurrence" refers to the average timing of the dominant seasonal month, whether we are discussing soil wetness or GPP. Specifically, for soil wetness, it denotes the average time of occurrence for the dominant wetness season, whereas for GPP, it denotes the average time of occurrence of the dominant GPP season. This have been discussed under the "Circularity statistics" subsection in Section 3.2. See Lines 180-183.

---

## Author Response (AR3)

**Editor**

Dear Authors,

Thank you for your revision. But some comments from reviewers are still not well addressed. For example, the last question from Anonymous referee #4 is not included in the main text. One more thing I'm still not very satisfied is the use of "soil water availability" in the text. Many new studies found that vegetation uptakes water not only from soil, but from bedrock fissures and groundwater (see https://www.nature.com/articles/s41586-021-03761-3; https://www.nature.com/articles/srep44110). Using "root zone water availability" is likely a more proper term (see https://hess.copernicus.org/articles/28/4477/2024/).
With these corrections, I think this manuscript will be well shaped up.

Thank you for your valuable feedback. We appreciate your attention to detail and the opportunity to further improve our manuscript. We have addressed both of your concerns.

**Comment:** On the use of "root zone water availability" instead of "soil water availability":

**Response:** We fully agree that "root zone water availability" is a more accurate and comprehensive term. We have updated all instances of "soil water availability" to "root zone water availability" throughout the manuscript and added supporting text in the introduction:

*Line 17 – 21: "Recent studies have shown that vegetation water uptake occurs not only from soil layers but also from bedrock fissures and groundwater systems (McCormick et al., 2021; Evaristo and McDonnell, 2017; Gao et al., 2024), suggesting that root zone water availability, rather than just soil wetness, determines the volume of water that plants can hydraulically lift ..."*

*Line 39-41: "Additionally, a global meta-analysis has shown that groundwater use by vegetation is widespread, with a global prevalence of 37\%, further emphasizing the importance of subsurface water sources for plant productivity (Evaristo and McDonnell, 2017)."*

*Line 48-51: "Recent research has emphasized the need for a more holistic understanding of the root zone in the Earth system, integrating its role across multiple spheres including the biosphere, hydrosphere, and atmosphere (Gao et al., 2024)."*

**Comment:** Addressing the last question from Anonymous Referee #4:

**Response:** While our original manuscript included discussion of average time of occurrence in Section 2.2 under circularity statistics, we acknowledge this could be clearer. We have now added an explicit clarification (lines 195-196):

*"The average time of occurrence represents the time of year when the flux (such as gross primary production, wetness, or vapor pressure deficit) typically reaches its peak, weighted by its intensity across all months."*